# Study on Resistant Hierarchical Fuzzy Neural Networks

Fengyu Gao [1,2], Jer-Guang Hsieh [1], Ying-Sheng Kuo [3] and Jyh-Horng Jeng [4,*]

[1] Department of Electrical Engineering, I-Shou University, Kaohsiung 84001, Taiwan; isu10702050d@cloud.isu.edu.tw (F.G.); jghsieh@isu.edu.tw (J.-G.H.)
[2] School of Electronic and Mechanical Engineering, Fujian Polytechnic Normal University, Fuqing 350300, China
[3] Department of Technology Management, Open University of Kaohsiung, Kaohsiung 81208, Taiwan; ysk@ouk.edu.tw
[4] Department of Information Engineering, I-Shou University, Kaohsiung 84001, Taiwan
[*] Correspondence: jjeng@isu.edu.tw

**Abstract:** Novel resistant hierarchical fuzzy neural networks are proposed in this study and their deep learning problems are investigated. These fuzzy neural networks can be used to model complex controlled plants and can also be used as fuzzy controllers. In general, real-world data are usually contaminated by outliers. These outliers may have undesirable or unpredictable influences on the final learning machines. The correlations between the target and each of the predictors are utilized to partition input variables into groups so that each group becomes the input variables of a fuzzy system in each level of the hierarchical fuzzy neural network. In order to enhance the resistance of the learning machines, we use the least trimmed squared error as the cost function. To test the resistance of learning machines to adverse effects of outliers, we add at the output node some noise from three different types of distributions, namely, normal, Laplace, and uniform distributions. Real-world datasets are used to compare the performances of the proposed resistant hierarchical fuzzy neural networks, resistant densely connected artificial neural networks, and densely connected artificial neural networks without noise.

**Keywords:** fuzzy neural network; hierarchical fuzzy neural network; outlier; resistant learning machine; deep learning



## 1. Introduction

No matter how the data are collected, the data at hand usually contain outliers. These data points are well separated from the bulk of data points or deviate from the general pattern of the data in some fashion. The outliers may have adverse or unpredictable influence on the final discriminant or predictive functions. In the past, many methods were proposed in statistical regression to address the problems with the outliers [1–6]. Regression is one of the major tasks in machine learning [7–10], and it is extensively studied using various models. Regression is applied in science education, agriculture, and signal processing [11–14]. The purpose of regression is to find the relationship between input variables and output variables in a dataset. However, the presence of noise and outliers changes the relationship. The main spirit of resistant regression is not to completely discard the outliers in the dataset, but to reduce the influence of these outliers on the final estimator. These robust regression problems were also investigated in the machine learning field [15–19]. The resistant regressors using the least trimmed squares (LTS) approach is particularly notable because of its simplicity and ease of use.

Fuzzy neural networks (FNNs) possess the advantages of both fuzzy systems [20,21] and neural networks [22,23], do not require accurate mathematical models, and have good learning ability which can approximate a wide range of nonlinear functions. FNNs have been widely used as machine learning models to deal with regression problems [24,25].

Next, we give a brief introduction to hierarchical fuzzy systems (HFSs) [26–30]. They made their debut in the control field and were devised to solve the problem of the curse of dimensionality (i.e., rule-explosion problem) in a fuzzy system. If a hierarchical fuzzy system is represented as network architecture, it is called a hierarchical fuzzy neural network (HFNN).

The main idea of HFSs is to appropriately partition input variables into groups so that each group becomes inputs of a low-dimensional fuzzy system. Each lower-dimensional fuzzy system is called a level in a hierarchical fuzzy system. Consider a simple hierarchical fuzzy neural network as shown in Figure 1. It has five inputs, a single output, and three levels. In this study, each low-dimensional fuzzy system will be replaced by an augmented fuzzy neural network (AFNN). Unlike the standard fuzzy network which has only a single hidden layer, an AFNN can have many hidden layers so that deep learning techniques can readily be applied to machine learning problems. Moreover, to enhance the representing power of the HFNN, each low-dimensional fuzzy system may have more than one output.

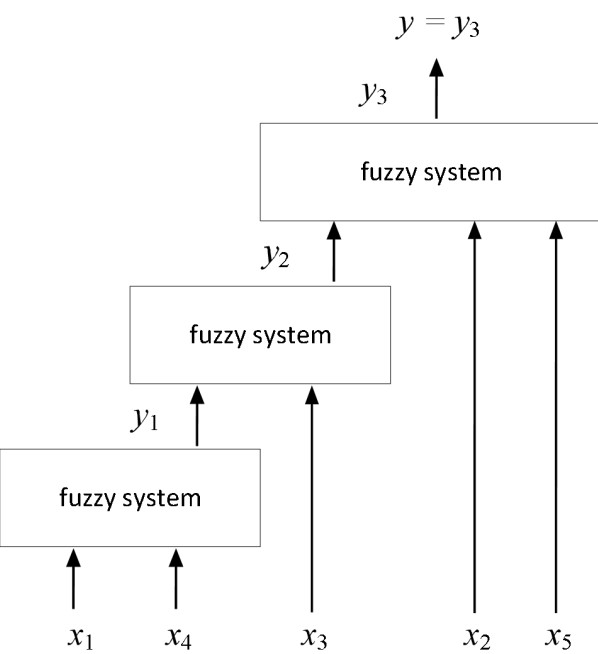

**Figure 1.** A simple hierarchical fuzzy neural network.

To the best of our knowledge, hierarchical fuzzy neural networks have never been used as resistant learning machines in past research. Furthermore, there have only been a few studies on traditional hierarchical fuzzy systems [28–30]. The authors proved the universal approximation property of the hierarchical fuzzy systems. Roughly speaking, this means that given a function $g$ with some kind of smoothness property and a positive constant $\epsilon$, no matter how small, there is a hierarchical fuzzy system $f$ (treated as a crisp nonlinear map), with the number of fuzzy rules unrestricted, such that the maximum absolute difference between $g$ and $f$ on a given compact set is less than $\epsilon$. Of course, this is an important property for any class of learning machines in successful application to real problems. It should be noted that the low-dimensional fuzzy system in each level is a Takagi–Sugeno–Kang (TSK) fuzzy system and the THEN parts of the fuzzy rules are all crisp polynomial functions but not the usual linear functions in TSK models. In [30], the authors developed a class of special hierarchical fuzzy systems where the outputs of the previous layer are not used in the IF parts, but used only in the THEN parts of the fuzzy rules of the current layer. This class of special hierarchical fuzzy systems also has the universal approximation property. The goal of this study is to investigate the performance of the proposed resistant hierarchical fuzzy neural networks against the adverse effect induced by the outliers.

## 2. Fuzzy Neural Networks

Consider a standard fuzzy system with $n$ inputs and $p$ outputs as shown in Figure 1 in [19]. Suppose the $m$ fuzzy rules in the rule base are given in canonical form as

$R_j$: IF $x_1$ is $A_{1j}$ and $x_2$ is $A_{2j}$ and ... and $x_n$ is $A_{nj}$,

THEN $y_1$ is $B_{j1}$ and $y_2$ is $B_{j2}$ and ... and $y_p$ is $B_{jp}$,

where $j \in \underline{m} = \{1, 2, ..., m\}$. The input–output mapping of the fuzzy system with a singleton fuzzifier, product inference engine, center-average defuzzifier, and Gaussian membership functions for the fuzzy sets $A_{ij}$ can be written as [19]:

$$y_k = \frac{\sum\limits_{j=1}^{m} w_{jk} \exp\left[-\sum\limits_{i=1}^{n} (x_i - c_{ij})^2 \Big/ \sigma_{ij}^2\right]}{\sum\limits_{j=1}^{m} \exp\left[-\sum\limits_{i=1}^{n} (x_i - c_{ij})^2 \Big/ \sigma_{ij}^2\right]}, k \in \underline{p},$$

$$x = \begin{bmatrix} x_1 & ... & x_n \end{bmatrix}^T \in \Re^n,$$

(1)

where, for $i \in \underline{n}$, $j \in \underline{m}$, and $k \in \underline{p}$,

$w_{jk}$: center of the normal fuzzy set $B_{jk}$

$c_{ij}$: center of the Gaussian fuzzy set $A_{ij}$

$\sigma_{ij}^2$: "variance" of the Gaussian fuzzy set $A_{ij}$

In order to escape from the division by zero in (1), to satisfy the constraints $\sigma_{ij}^2 > 0$, and to allow the learning machine to handle broader types of data, a better parameterization is given by

$$y_k = f_{ok}(x) = f_{ok}\left(\frac{\sum\limits_{j=1}^{m} w_{jk} \exp\left[-\sum\limits_{i=1}^{n} (x_i - c_{ij})^2 \exp(v_{ij})\right]}{\sum\limits_{j=1}^{m} \exp\left[-\sum\limits_{i=1}^{n} (x_i - c_{ij})^2 \exp(v_{ij})\right]}\right),$$

$$k \in \underline{p},$$

(2)

where $v_{ij} = \log\left(1\Big/\sigma_{ij}^2\right)$ and $f_{ok}$ is the activation function of the $k$th output node.

For $i \in \underline{n}$, $j \in \underline{m}$, and $k \in \underline{p}$, we define

$$u_j = \sum_{i=1}^{n} (x_i - c_{ij})^2 \exp(v_{ij}), r_j = \exp(-u_j),$$

(3a)

$$s_k = \sum_{j=1}^{m} w_{jk} r_j, g = \sum_{j=1}^{m} r_j,$$

(3b)

then

$$y_k = f_{ok}(s_k/g).$$

(3c)

According to (3), the fuzzy system given in (2) can now be represented as a feedforward neural network [19], called a fuzzy neural network (FNN), as shown schematically in Figure 2. Note that there is only a single hidden layer in this neural network.

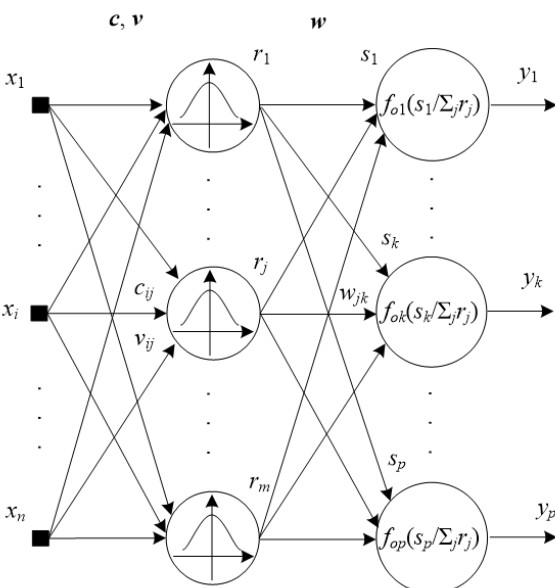

**Figure 2.** Fuzzy neural network with a single hidden layer.

As mentioned earlier, to enhance the learning capability or predictive power of the proposed hierarchical fuzzy neural network model, it would be a good idea to allow many hidden layers in a fuzzy neural network. In that case, we call it an augmented fuzzy neural network (AFNN), which is shown schematically in Figure 3. In the hierarchical fuzzy neural networks studied here, each low-dimensional fuzzy system will be replaced by an AFNN.

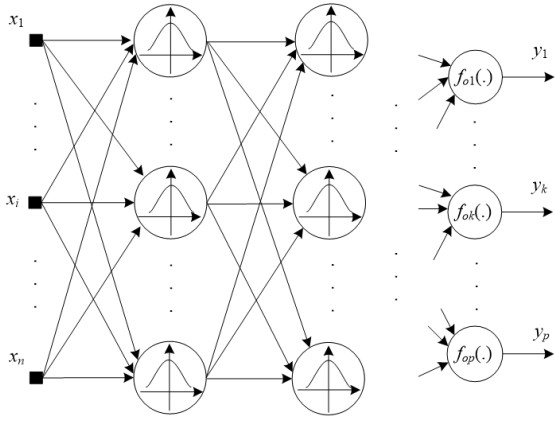

**Figure 3.** Augmented fuzzy neural network.

## 3. Loss Function

There are many choices for the loss functions in the resistant machine learning problem. In the following development, we will use the least trimmed squares (LTS) loss function. The methodology by using other loss functions is similar and will not be repeated here. Now, we briefly review the LTS loss function.

Let $X \subseteq \Re^n$ and $Y \subseteq \Re$. Suppose the training data are given by

$$S := \left\{ (x_q, d_q) \right\}_{q=1}^l \subseteq X \times Y.$$

Let $y_q$, $q \in \underline{l}$, denote the predicted response value corresponding to the predictor $x_q$. Then, its residual is defined by $e_q = d_q - y_q$. The LTS approach to resistant regression is to search for the connection weights $c_{ij}$, $v_{ij}$, $w_{jk}$ of the neural network so as to minimize the following loss function:

$$J_{lts} = \sum_{q=1}^{l} a\left[R\left(e_q^2\right)\right]e_q^2,$$

where the score function $a[\cdot]$ (i.e., penalizing weight) is defined by

$$a_q := \begin{cases} 1, & 1 \le q \le l*, \\ 0, & l* < q \le l, \end{cases}$$

and $R\left(e_q^2\right)$ is the rank (from the smallest to the largest) of $e_q^2$ in $e_1^2, \ldots, e_l^2$. We call $h = (l - l*)/l$ the trimming percentage. It is the most important design parameter in LTS regression.

### 4. Partition of Input Variables

The biggest problem in using a hierarchical fuzzy neural network as a learning machine is how to appropriately partition the input variables so that each group becomes inputs of a low-dimensional fuzzy system. After several trials, we found that the correlation between the target and each of the predictors, i.e., correlation coefficient, may reliably be used to determine the input variables for the fuzzy neural network in each level of the hierarchical fuzzy neural network under consideration. Once this is done, the following rules of thumb may then be applied:

Rule 1: Correlation is measured in absolute value (i.e., magnitude).
Rule 2: Put the more correlated predictors in our hierarchical fuzzy neural network as early as possible.
Rule 3: Variables with low correlation will not be used as predictors.
Rule 4: Predictors with about the same level of correlation may be collected in the same group.

### 5. Illustrative Examples

To better evaluate the performances of the resistant hierarchical fuzzy neural network, the $k$-fold cross validation, one of the commonly used cross validation techniques, will be employed. First, the given data are randomly partitioned into $k$ parts which are roughly equal in size. In each run of the method, one part is left as a validating set with the remaining used as the training set. After training with the training set, the pre-specified performance indicators of the validating set are computed. Those $k$ values of the performance indicators from the $k$ runs are then averaged to give estimates of the performance indicators. In this study, we use 10-fold cross validation (i.e., $k = 10$) and we take the median to average the values of the performance indicators. The numerical variables in the dataset are standardized before cross validation.

To test the robustness or resistance of the learning machine, we will add three kinds of noise to the output of the training data. These are random samples drawn from the following probability distributions:

(1) normal distribution: $\mu = 0, \sigma = 1$.
$$f(x) = \frac{1}{\sigma\sqrt{2\pi}} \exp\left[-\frac{1}{2}\left(\frac{x-\mu}{\sigma}\right)^2\right], -\infty < x < \infty.$$

(2) Laplace distribution: $\mu = 0, b = 1$.
$$f(x) = \frac{1}{2b} \exp\left(-\frac{|x-\mu|}{b}\right), -\infty < x < \infty \ (b > 0).$$

(3) Uniform distribution: $a = -1, b = 1$.
$$f(x) = \begin{cases} 1/(b-a), & a \le x \le b, \\ 0, & otherwise. \end{cases}$$

The noise will not be added to the validating set.

In the following simulations using LTS loss function, we set the trimming percentage to be 0.2.

One natural question is that can HFNNs compete with the state-of-the-art artificial neural networks, e.g., densely connected artificial neural networks (DNNs)? Ten real-world datasets described in Table 1 are used in this study to compare the performances of the

LTS-HFNNs, LTS-DNNs, and DNNs without noise. These datasets can be loaded from the UCI machine learning repository of the University of California at Irvine [31], LIBSVM of National Taiwan University [32], and Kaggle [33].

For example, the first dataset shown in Table 1 is "Airfoil Self-Noise" from NASA in the benchmark UCI. It is obtained from a series of aerodynamic and acoustic tests of two- and three-dimensional airfoil blade sections conducted in an anechoic wind tunnel. The number of instances (records) is 1503 and the number of attributes is 6. The five input predictors are frequency, angle of attack, chord length, free-stream velocity, and suction side displacement thickness. The only output is the scaled sound pressure level. In other words, we are performing a 5-input/1-output regression task for this dataset. All other datasets can be described similarly.

**Table 1.** Description of the datasets.

| Dataset | No. of Cases | No. of Predictors | Source |
|---|---|---|---|
| Airfoil self-noise | 1503 | 5 | UCI |
| Boston housing | 506 | 13 | Kaggle |
| Combined cycle power plant | 9568 | 4 | UCI |
| Concrete compressive strength | 1030 | 8 | UCI |
| Cpusmall | 8192 | 12 | LIBSVM |
| Mg | 1385 | 6 | LIBSVM |
| Parkinsons telemonitoring (motor UPDRS) | 5875 | 16 | UCI |
| Parkinsons telemonitoring (total UPDRS) | 5875 | 16 | UCI |
| QSAR fish toxicity | 908 | 6 | UCI |
| Space-GA | 3107 | 6 | LIBSVM |

Next, we list the architecture of the HFNNs and DNNs, as shown in Table 2. For fair comparison, the DNN in each of the following simulations is configured such that the number of adjustable parameters of the DNN is about the same as that of the corresponding HFNN. The integers in the parentheses of the second and third columns are the number of adjustable parameters of the neural networks.

**Example 1.** *First, we consider adding the output noise from the normal distribution. Table 3 shows the 10-fold cross validated simulation results, where the values in parentheses are the standard errors (i.e., standard deviations) of the performance indicators. In the simulations, three performance indicators are used. In Table 3, "LOSS" denotes the value of the LTS loss function, "MSE" is the mean squared error, and "MAE" is the mean absolute error. In the last column of the table, we list the simulation results for DNNs without noise, where the loss function is the same as MSE. Those values serve as goal values for LTS-HFNNs and LTS-DNNs. As observed in Table 3, for the first dataset "Airfoil self-noise", the MSE of LTS-HFNN is 0.2970 which is roughly the same as that of LTS-DNN, 0.2916. Both are higher than that of DNN, 0.1039, since in this method there are no Gaussian noises added on the output variable. One important statistic is the standard error of the 10-fold cross validation. Those values are 0.0600, 0.3971, and 0.4019. Smaller standard errors reveal more stable estimations. In fact, except for the "Boston housing" data, the proposed LTS-HFNNs have smaller standard errors for the remaining 9 datasets. Roughly speaking, the proposed LTS-HFNNs usually have slightly higher mean values of MSE and MAE, but smaller standard errors.*

**Table 2.** Architecture of HFNNs and DNNs.

| Dataset | HFNN | DNN |
|---|---|---|
| Airfoil self-noise | (1,4,3,2)<br>(4,4,3,2)<br>(4,4,3,1)<br>(159) | 5-10-6-4-1<br>(159) |
| Boston housing | (3,4,3,2)<br>(5,4,3,2)<br>(6,4,3,1)<br>(5,4,3,1)<br>(269) | 13-10-7-5-1<br>(263) |
| Combined cycle power plant | (1,4,3,2)<br>(3,4,3,2)<br>(4,4,3,1)<br>(151) | 4-8-8-4-1<br>(153) |
| Concrete compressive strength | (2,4,3,2)<br>(5,4,3,2)<br>(5,4,3,1)<br>(183) | 8-8-7-5-1<br>(181) |
| Cpusmall | (2,4,3,2)<br>(6,4,3,2)<br>(6,4,3,2)<br>(4,4,3,1)<br>(261) | 12-10-8-4-1<br>(259) |
| Mg | (2,4,3,2)<br>(4,4,3,2)<br>(4,4,3,1)<br>(167) | 6-8-7-5-1<br>(165) |
| Parkinsons telemonitoring (motor UPDRS) | (2,4,3,2)<br>(4,4,3,2)<br>(5,4,3,1)<br>(175) | 16-6-5-5-1<br>(173) |
| Parkinsons telemonitoring (total UPDRS) | (1,4,3,2)<br>(4,4,3,2)<br>(4,4,3,1)<br>(159) | 16-6-5-3-1<br>(159) |
| QSAR fish toxicity | (1,4,3,2)<br>(4,4,3,2)<br>(3,4,3,2)<br>(4,4,3,1)<br>(213) | 6-9-8-7-1<br>(214) |
| Space-GA | (1,4,3,2)<br>(3,4,3,2)<br>(5,4,3,2)<br>(3,4,3,1)<br>(213) | 6-9-8-7-1<br>(214) |

**Table 3.** Simulation results in Example 1.

| | LTS-HFNN | LTS-DNN | DNN without Noise |
|---|---|---|---|
| **Airfoil self-noise** | | | |
| LOSS | 0.0919 (±0.0201) | 0.0901 (±0.1667) | |
| MSE | 0.2970 (±0.0600) | 0.2916 (±0.3971) | 0.1039 (±0.4019) |
| MAE | 0.4118 (±0.0410) | 0.4022 (±0.2255) | 0.2374 (±0.2563) |
| **Boston housing** | | | |
| LOSS | 0.3005 (±0.0691) | 0.2220 (±0.0500) | |
| MSE | 0.9877 (±0.3232) | 0.5845 (±0.2183) | 0.1017 (±0.0981) |
| MAE | 0.7745 (±0.0841) | 0.5928 (±0.0817) | 0.2359 (±0.0581) |
| **Combined cycle power plant** | | | |
| LOSS | 0.0260 (±0.0036) | 0.0275 (±0.1481) | |
| MSE | 0.0734 (±0.0107) | 0.0744 (±0.2897) | 0.0570 (±0.4011) |
| MAE | 0.2133 (±0.0124) | 0.2144 (±0.2031) | 0.1850 (±0.2910) |
| **Concrete compressive strength** | | | |
| LOSS | 0.1481 (±0.0352) | 0.1947 (±0.1487) | |
| MSE | 0.3964 (±0.0743) | 0.4904 (±0.3362) | 0.1393 (±0.2216) |
| MAE | 0.4986 (±0.0478) | 0.5574 (±0.1814) | 0.2746 (±0.1489) |
| **Cpusmall** | | | |
| LOSS | 0.0330 (±0.0048) | 0.0327 (±0.0248) | |
| MSE | 0.8737 (±0.1438) | 0.1296 (±0.4685) | 0.0253 (±0.0035) |
| MAE | 0.3928 (±0.0343) | 0.2518 (±0.1383) | 0.1141 (±0.0062) |
| **Mg** | | | |
| LOSS | 0.1538 (±0.0211) | 0.1507 (±0.1192) | |
| MSE | 0.4595 (±0.0747) | 0.4139 (±0.2331) | 0.2920 (±0.2131) |
| MAE | 0.5262 (±0.0435) | 0.5047 (±0.1335) | 0.4157 (±0.1297) |
| **Parkinsons telemonitoring (motor UPDRS)** | | | |
| LOSS | 0.3616 (±0.0319) | 0.4102 (±0.0979) | |
| MSE | 0.9389 (±0.0684) | 0.9262 (±0.1149) | 0.7059 (±0.0996) |
| MAE | 0.7757 (±0.0329) | 0.7952 (±0.0801) | 0.6746 (±0.0637) |
| **Parkinsons telemonitoring (total UPDRS)** | | | |
| LOSS | 0.3333 (±0.0404) | 0.3729 (±0.0578) | |
| MSE | 0.9158 (±0.0686) | 0.9454 (±0.0937) | 0.7011 (±0.1500) |
| MAE | 0.7536 (±0.0393) | 0.7744 (±0.0501) | 0.6689 (±0.0728) |
| **QSAR fish toxicity** | | | |
| LOSS | 0.2175 (±0.0419) | 0.2016 (±0.0671) | |
| MSE | 0.6589 (±0.1362) | 0.6178 (±0.2001) | 0.4719 (±0.1418) |
| MAE | 0.6108 (±0.0463) | 0.5904 (±0.0793) | 0.4779 (±0.0923) |
| **Space-GA** | | | |
| LOSS | 0.1474 (±0.0276) | 0.1145 (±0.0623) | |
| MSE | 0.5002 (±0.1766) | 0.3512 (±0.1806) | 0.2716 (±0.2544) |
| MAE | 0.5074 (±0.0478) | 0.4469 (±0.0890) | 0.3911 (±0.1295) |

**Example 2.** *Next, we consider adding the output noise from the Laplace distribution. Table 4 shows the 10-fold cross validated simulation results. As observed in Table 4, the LOSS and MSE of LTS-HFNN for 4 out of 10 datasets are slightly smaller than for LTS-DNN, and for the remaining 6 datasets they are slightly higher than for LTS-DNN. The MAE of LTS-HFNN for 3 out of 10 datasets is smaller than for LTS-DNN, and for the remaining 7 datasets it is higher than for LTS-DNN. For all LOSS, MSE, and MAE values, the standard errors of LTS-HFNN are obviously much smaller than for LTS-DNN for all 10 datasets, except for MAE in "Boston housing". In general, LTS-DNNs have slightly smaller mean values of MSE and MAE, but larger standard errors.*

**Table 4.** Simulation results in Example 2.

|  | LTS-HFNN | LTS-DNN | DNN without Noise |
|---|---|---|---|
| **Airfoil self-noise** | | | |
| LOSS | 0.1096 (±0.0273) | 0.2661 (±0.1869) | |
| MSE | 0.3451 (±0.0621) | 0.6195 (±0.4015) | 0.1039 (±0.4019) |
| MAE | 0.4478 (±0.0478) | 0.6258 (±0.2294) | 0.2374 (±0.2563) |
| **Boston housing** | | | |
| LOSS | 0.3600 (±0.1021) | 0.2086 (±0.0765) | |
| MSE | 1.0135 (±0.2562) | 0.8298 (±0.3050) | 0.1017 (±0.0981) |
| MAE | 0.7595 (±0.0897) | 0.6845 (±0.0892) | 0.2359 (±0.0581) |
| **Combined cycle power plant** | | | |
| LOSS | 0.0247 (±0.0016) | 0.0265 (±0.1501) | |
| MSE | 0.0703 (±0.0057) | 0.0711 (±0.2893) | 0.0570 (±0.4011) |
| MAE | 0.2041 (±0.0061) | 0.2103 (±0.2067) | 0.1850 (±0.2910) |
| **Concrete compressive strength** | | | |
| LOSS | 0.1658 (±0.0425) | 0.1507 (±0.1461) | |
| MSE | 0.4358 (±0.0972) | 0.4265 (±0.3083) | 0.1393 (±0.2216) |
| MAE | 0.5256 (±0.0569) | 0.5078 (±0.1754) | 0.2746 (±0.1489) |
| **Cpusmall** | | | |
| LOSS | 0.0205 (±0.0063) | 0.0217 (±0.0279) | |
| MSE | 0.4521 (±0.2958) | 0.0929 (±0.4073) | 0.0253 (±0.0035) |
| MAE | 0.2949 (±0.0819) | 0.2052 (±0.1373) | 0.1141 (±0.0062) |
| **Mg** | | | |
| LOSS | 0.1449 (±0.0367) | 0.1374 (±0.1370) | |
| MSE | 0.4620 (±0.0654) | 0.3965 (±0.2542) | 0.2920 (±0.2131) |
| MAE | 0.5013 (±0.0443) | 0.4823 (±0.1512) | 0.4157 (±0.1297) |
| **Parkinsons telemonitoring (motor UPDRS)** | | | |
| LOSS | 0.3612 (±0.0380) | 0.3407 (±0.0875) | |
| MSE | 0.9004 (±0.0912) | 0.8740 (±0.1118) | 0.7059 (±0.0996) |
| MAE | 0.7605 (±0.0387) | 0.7441 (±0.0763) | 0.6746 (±0.0637) |
| **Parkinsons telemonitoring (total UPDRS)** | | | |
| LOSS | 0.3287 (±0.0300) | 0.3946 (±0.0578) | |
| MSE | 0.8618 (±0.0619) | 0.9758 (±0.0921) | 0.7011 (±0.1500) |
| MAE | 0.7409 (±0.0305) | 0.7943 (±0.0489) | 0.6689 (±0.0728) |
| **QSAR fish toxicity** | | | |
| LOSS | 0.2311 (±0.0651) | 0.1886 (±0.1202) | |
| MSE | 0.7317 (±0.1558) | 0.5801 (±0.2875) | 0.4719 (±0.1418) |
| MAE | 0.6655 (±0.0762) | 0.5735 (±0.1287) | 0.4779 (±0.0923) |
| **Space-GA** | | | |
| LOSS | 0.1379 (±0.0077) | 0.1069 (±0.0862) | |
| MSE | 0.4689 (±0.1199) | 0.3195 (±0.2381) | 0.2716 (±0.2544) |
| MAE | 0.5055 (±0.0152) | 0.4410 (±0.1197) | 0.3911 (±0.1295) |

**Example 3.** *Finally, we consider adding the output noise from the uniform distribution. Table 5 shows the 10-fold cross validated simulation results. As observed in Table 5, the LOSS of LTS-HFNN for 6 out of 10 datasets are smaller than for LTS-DNN, and for the remaining 4 datasets it is higher than for LTS-DNN. The MSE and MAE of LTS-HFNN for 8 out of 10 datasets are smaller than for LTS-DNN, and for the remaining 2 datasets they are higher than for LTS-DNN. As in the previous two examples, LTS-FNNs usually have slightly higher mean values of MSE and MAE, but smaller standard errors.*

**Table 5.** Simulation results in Example 3.

|  | LTS-HFNN | LTS-DNN | DNN without Noise |
|---|---|---|---|
| Airfoil self-noise | | | |
| LOSS | 0.0878 (±0.0127) | 0.0801 (±0.1175) | |
| MSE | 0.2787 (±0.0543) | 0.2720 (±0.2748) | 0.1039 (±0.4019) |
| MAE | 0.3961 (±0.0273) | 0.3829 (±0.1545) | 0.2374 (±0.2563) |
| Boston housing | | | |
| LOSS | 0.1799 (±0.0378) | 0.1248 (±0.0521) | |
| MSE | 0.6684 (±0.2669) | 0.4135 (±0.1432) | 0.1017 (±0.0981) |
| MAE | 0.5968 (±0.0777) | 0.4861 (±0.0773) | 0.2359 (±0.0581) |
| Combined cycle power plant | | | |
| LOSS | 0.0271 (±0.0019) | 0.0295 (±0.1466) | |
| MSE | 0.0761 (±0.0063) | 0.0780 (±0.3029) | 0.0570 (±0.4011) |
| MAE | 0.2155 (±0.0072) | 0.2185 (±0.2044) | 0.1850 (±0.2910) |
| Concrete compressive strength | | | |
| LOSS | 0.1160 (±0.0262) | 0.0897 (±0.1117) | |
| MSE | 0.3350 (±0.0706) | 0.2479 (±0.2392) | 0.1393 (±0.2216) |
| MAE | 0.4414 (±0.0468) | 0.3816 (±0.1371) | 0.2746 (±0.1489) |
| Cpusmall | | | |
| LOSS | 0.0267 (±0.0062) | 0.0311 (±0.0157) | |
| MSE | 0.8325 (±0.0999) | 0.1182 (±0.3180) | 0.0253 (±0.0035) |
| MAE | 0.3724 (±0.0357) | 0.2518 (±0.0899) | 0.1141 (±0.0062) |
| Mg | | | |
| LOSS | 0.1213 (±0.0249) | 0.1249 (±0.1376) | |
| MSE | 0.3826 (±0.0528) | 0.3638 (±0.2730) | 0.2920 (±0.2131) |
| MAE | 0.4662 (±0.0338) | 0.4643 (±0.1566) | 0.4157 (±0.1297) |
| Parkinsons telemonitoring (motor UPDRS) | | | |
| LOSS | 0.3412 (±0.0275) | 0.2980 (±0.0684) | |
| MSE | 0.9623 (±0.0748) | 0.8299 (±0.0921) | 0.7059 (±0.0996) |
| MAE | 0.7576 (±0.0291) | 0.7097 (±0.0522) | 0.6746 (±0.0637) |
| Parkinsons telemonitoring (total UPDRS) | | | |
| LOSS | 0.3184 (±0.0318) | 0.3892 (±0.0752) | |
| MSE | 0.9121 (±0.0766) | 0.9835 (±0.1188) | 0.7011 (±0.1500) |
| MAE | 0.7326 (±0.0324) | 0.7991 (±0.0644) | 0.6689 (±0.0728) |
| QSAR fish toxicity | | | |
| LOSS | 0.1507 (±0.0441) | 0.1246 (±0.0489) | |
| MSE | 0.4986 (±0.1436) | 0.4540 (±0.1513) | 0.4719 (±0.1418) |
| MAE | 0.5347 (±0.0696) | 0.5097 (±0.0838) | 0.4779 (±0.0923) |
| Space-GA | | | |
| LOSS | 0.1224 (±0.0073) | 0.0980 (±0.0114) | |
| MSE | 0.4284 (±0.0993) | 0.2986 (±0.0790) | 0.2716 (±0.2544) |
| MAE | 0.4792 (±0.0176) | 0.4196 (±0.0284) | 0.3911 (±0.1295) |

In the above three examples, we have shown the robustness of the proposed FNNs through adding various types of noise to the datasets. In the simulations, we conducted cross validations to calculate the means and standard deviations of the testing errors as the performance indicators. As observed, the proposed LTS-HFNN and LTS-DNN both exhibited very good robustness.

## 6. Conclusions

In this study, we proposed resistant hierarchical fuzzy neural networks and investigated the associated deep learning problems. Correlations between the target and each of the predictors, together with some rules of thumb, were utilized to partition input variables into groups so that each group becomes the input variables of a fuzzy system in each level of the hierarchical fuzzy neural network. The least trimmed squared error was used

as the cost function in order to enhance the resistance of the hierarchical fuzzy neural networks. Three different types of noise were added to the output node of the neural network to test the resistance of the neural networks. From the simulation results, it seems that the performance has very little connection with the types of the noise. The proposed LTS-HFNN was compared with LTS-DNN, and DNN without noises was also taken for comparison. Ten datasets were utilized for regression. Experimental results shows that the added noises for LTS-HFNN and LTS-DNN increase the loss. Furthermore, LTS-FNNs usually have slightly higher mean values of MSE and MAE, but smaller standard errors. Smaller standard errors correspond to stronger robustness of the models. Sacrificing a small accuracy for much stronger robustness is worth the cost when real-world datasets have a certain number of outliers. Because of the proposed hierarchical structure together with the augmented fuzzy neural units and the least trimmed squared error, robust fuzzy neural networks are obtained in our works.

The most important future work of hierarchical fuzzy neural networks is to develop more powerful techniques to partition the input variables. One immediate technique we can think of is the decision tree. It may help us to appropriately partition the input variables in a particular way. Moreover, the input variables used in earlier levels can also be used in later levels. Another interesting topic is to investigate the resistance of hierarchical fuzzy neural networks against the $x$-space outliers. These $x$-space outliers can be generated by adding various types of noises in the input variables.

**Author Contributions:** Conceptualization, J.-G.H.; methodology, J.-G.H. and F.G.; validation, F.G. and J.-H.J.; formal analysis, J.-G.H. and J.-H.J.; writing—original draft preparation, Y.-S.K.; writing—review and editing, F.G., J.-G.H., J.-H.J. and Y.-S.K.; visualization, F.G.; supervision, J.-H.J. and J.-G.H.; project administration, J.-G.H. and J.-H.J. All authors have read and agreed to the published version of the manuscript.

**Funding:** This research received no external funding.

**Acknowledgments:** The authors acknowledge the Ministry of Science and Technology, Taiwan, through grant number MOST-109-2221-E-214-013 and MOST 110-2221-E-214-019.

**Conflicts of Interest:** The authors declare no conflict of interest.

## Nomenclature

| | |
|---|---|
| $\underline{p}$ | $\underline{p} := \{1, 2, \cdots, p\}$ |
| $\Re^n$ | $n$-dimensional real space |
| $X \times Y$ | Cartesian product of sets $X$ and $Y$ |

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
