# Peer review of "Study on Resistant Hierarchical Fuzzy Neural Networks"

_electronics, doi:10.3390/electronics11040598_

Round 1

Reviewer 1 Report

In their work, the authors presented resistant hierarchical fuzzy neural networks and investigated the associated deep learning problems. In doing so, three different noise types were added to the output node of the neural network to test the resistance of the neural networks. However, from the simulation results it seems that the performance has very little connection with the types of the noise. Maybe I missed it, but why did you exactly chose these 10 datasets? How was the selection process? Some figures regarding the datasets would be nice, I don’t know all these datasets and what they “show”, e.g. are they all 2D or some 3D? At the end of the conclusion should be more statements about future work.

Reviewer 2 Report

  1. Figure 2. Fuzzy neural network. edit the caption
    2. Cite the relevant references in all the section
  2. 3. After 3 examples please mention a paragraph of comparison or conclude the results.
  3. There must be more references related to the literature review. 
  4. Improve the introduction and literature review part.

Round 2

Reviewer 1 Report

Thanks for addressing my comments, I endorse the paper for publication.